# Progress and Applications of Seawater-Activated Batteries

**Jinmao Chen, Wanli Xu, Xudong Wang, Shasha Yang and Chunhua Xiong \***

Institute of System Engineering, Academy of Military Science, Beijing 102300, China
* Correspondence: energyxch@163.com

**Abstract:** Obtaining energy from renewable natural resources has attracted substantial attention owing to their abundance and sustainability. Seawater is a naturally available, abundant, and renewable resource that covers >70% of the Earth's surface. Reserve batteries may be activated by using seawater as a source of electrolytes. These batteries are very safe and offer a high power density, stable discharge voltage, high specific energy, and long dry storage life and are widely used in marine exploration instruments, life-saving equipment, and underwater weaponry. This review provides a comprehensive introduction to seawater-activated batteries. Here, we classify seawater-activated batteries into metal semi-fuel, high-power, and rechargeable batteries according to the different functions of seawater within them. The working principles and characteristics of these batteries are then introduced, and we describe their research statuses and practical applications. Finally, we provide an outlook on the development of seawater-activated batteries and highlight practical issues to drive further progress.

**Keywords:** seawater-activated battery; marine power sources; electrical energy storage; electrode materials

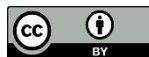

## 1. Introduction

Renewable energy technologies (e.g., solar, seawater, wind, and tidal) have been identified as potential alternatives to thermal and nuclear energy for producing green electricity [1–3], while innovations in energy capture have brought immense opportunities and challenges to the energy industry. Electricity is the backbone of modern industrial society and has a wide range of applications, including transport, communication, heating, and lighting. Traditionally, most electricity has been generated from the burning of non-renewable fossil fuels, which has resulted in the exhaustion of fuel reserves and global climate change [4,5]. The emergence of renewable energy technologies has changed the methods of electrical generation. Various renewable and clean energy sources, such as marine, solar, geothermal, and wind energy, have been developed as complementary or alternative technologies to mainstream fossil fuels [6–8].

Seawater-activated batteries are chemical power sources that use seawater as an electrolyte [9], with active metal as the negative electrode material; the positive electrode material is usually $AgCl$, $PbCl_2$, $CuCl_2$, other metal halides, or the oxygen dissolved in seawater [10,11]. The active metal of the negative electrode is dissolved in seawater, and a negative current is generated, while the positive metal halide or dissolved oxygen (DO) provides a positive current for the battery when reduction occurs. Dissolved oxygen seawater-activated batteries (DO-type seawater batteries) use magnesium as the anode, carbon as the cathode, seawater as the electrolyte, and dissolved oxygen in seawater as the oxidant. The battery is normally dry when not in use. Natural seawater is used as an electrolyte to ensure the directional movement of ions in the electrode reaction and to form a continuous and stable current [12–15].

The main factor affecting seawater-activated batteries is the choice of electrode materials that can determine the discharge voltage and the actual capacity of the battery.

Anodes are generally made of relatively low-priced metals, including lithium, sodium, magnesium, aluminum, zinc, etc., which theoretically provide higher electrical power [16–18]. However, the alkali metals lithium and sodium are highly reactive and react violently with seawater, which is difficult to control, therefore, they are less used [19]. Zinc is easy to process and generally does not have side reactions with seawater, but its voltage is much lower than that of magnesium, and the zinc dissolved in the discharge is easily reduced, forming dendritic zinc crystals, leading to short circuits in the battery [20]. Therefore, the anode materials for seawater-activated batteries mostly employ inexpensive and abundant magnesium and aluminum metals. However, the actual voltage of the seawater-activated battery with magnesium and aluminum as the anode is less than the theoretical voltage of the battery, mainly owing to the formation of a hydroxide passivation film on the anode surface, so, the electrode mainly uses alloys instead of pure metal [21].

Currently, magnesium alloy is mainly a five-element alloy, and aluminum alloy is mainly a six-element alloy [22]. Alloying is a common method to improve the electrochemical activity and corrosion resistance of electrodes. There are two main types of alloying elements under study: activating elements such as Ga, In, Tl, Mg, or Hg, which destroy the dense passivation film of the alloy anode, and corrosion-inhibiting elements such as Sn, Pb, Bi, Zn, or Hg, which have high hydrogen precipitation over potential [23]. In addition, the cathode materials for seawater-activated battery materials are AgCl, CuCl, $PbCl_2$, AgO, conductive polymers, and carbon fiber materials. Conductive polyaniline is used as cathode material for primary and secondary batteries as well as metal-air batteries due to its high conductivity, large specific surface area, good stability in oxygen and water, easy synthesis, convenient doping, strong charge storage capacity, and good electrochemical performance [24]. Carbon fiber materials used as cathodes for seawater-activated batteries have high impact resistance, light weight, and corrosion resistance [25].

The most prominent feature of seawater-activated batteries is that the electrolyte comes from the working environment of the battery, thus making it suitable in a variety of marine environments [26]. Compared with other types of chemical power sources, such as traditional (e.g., lead–acid [27], silver–zinc [28], and alkaline batteries [29]) and lithium batteries [30] and fuel cells [31], seawater-activated batteries [32] have received widespread attention because of their environmental adaptability and safety. As a newly developed power source, seawater-activated batteries use active metals as anodes and rely on their activation via dissolution in seawater to provide a current. Owing to their high energy density, long storage time, and safety, they have been widely used in military equipment [33,34].

The first seawater-activated batteries were Mg/AgCl batteries, which were already in use as power sources for electric torpedoes during World War II and were later expanded to power buoys, sounding balloons, beacon lights, life-saving equipment, and autonomous underwater vehicles (AUVs) (Figure 1). Owing to the high price of silver chloride, seawater-activated battery products have long been limited to military use. To promote the commercial and civil use of seawater-activated battery products, Mg/CuCl seawater-activated batteries were introduced in 1949, followed by $Mg/PbCl_2$, $Mg/HgCl_2$, $Mg/PbO_2$, and Al/AgO batteries. Continued research on seawater-activated batteries has led to the pairing of almost all positive and negative materials, and a variety of practical battery series with different structures and properties have been developed to meet the requirements of military and civil applications [35,36].

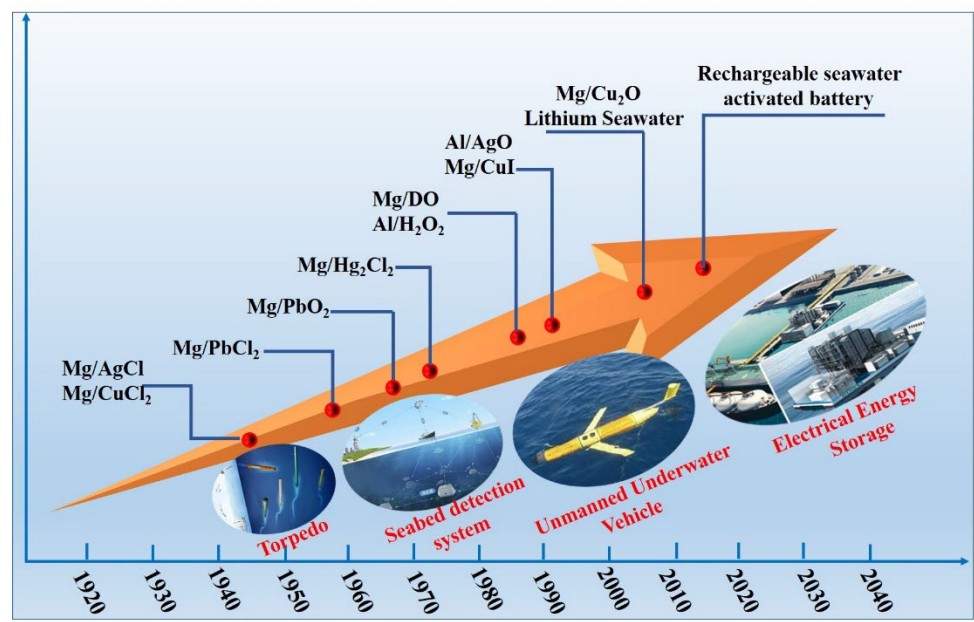

**Figure 1.** Timeline of the development of seawater-activated batteries and major innovations and changes, from a primary to a secondary system.

To meet the requirements for deep-sea applications or long-term undersea work, metal-seawater (anode-cathode) batteries, such as Mg–DO and Al–$H_2O_2$ seawater-activated batteries, were exploited for their high-energy densities at the end of the 20th century [37]. Dissolved oxygen seawater-activated batteries depend on the corrosion of an active metal anode and oxygen reduction of an inert cathode to produce a potential of approximately 1 V when they are immersed in seawater. Because oxygen is not easily dissolved in seawater, such batteries are characterized by low cathode current densities and are therefore best suited for long-term low-power applications [38]. The storage of electrical energy using primary seawater-activated batteries is impractical. Therefore, in 2014, Kim et al. [39] proposed and patented a rechargeable seawater-activated battery that uses seawater as a $Na^+$ ion cathode to store electrical energy. This rechargeable battery has good prospects for a wide range of applications, such as energy storage systems for tidal and wind power generation in coastal areas and military energy providers, among others. The performance comparison between different seawater-activated batteries is shown in Table 1.

**Table 1.** Performance comparison of different seawater-activated batteries.

| Type | Working Mechanism | Energy Density | Characteristic | Application |
|---|---|---|---|---|
| Dissolved oxygen seawater-activated batteries | Anode: $M \rightarrow M^{n+} + ne^-$<br><br>Cathode: $O_2 + 2H_2O + 4e^- \rightarrow 4OH^-$ | 50–150 Wh/kg | High specific energy;<br>Low output power;<br>Long storage life;<br>High security | deep sea survey operation |
| Metal hydrogen peroxide seawater-activated batteries | Anode: $M + nOH^- \rightarrow M(OH)n + ne^-$<br><br>Cathode: $H_2O_2 + e^- \rightarrow 2OH^-$ | 100–500 Wh/kg | High specific energy;<br>Long storage life;<br>High security | Torpedoes;<br>Unmanned Underwater Vehicle |

| High-power seawater-activated batteries | Anode:$Al + 4OH^- \rightarrow Al(OH)_4^{-1} + e^-$ <br> Cathode:$AgO + H_2O + 2e^- \rightarrow Ag_2O + 2OH^-$ <br> $Ag_2O + H_2O + 2e^- \rightarrow 2Ag + 2OH^-$ | 130 Wh/kg | High current discharge; High output power; Long storage life; High security | Torpedoes |
| | Anode: $Mg \rightarrow Mg^{2+} + 2e^-$ <br> Cathode: $AgCl + e^- \rightarrow Ag + Cl^-$ | 88 Wh/kg | Low output power; Long discharge time; Long storage life; High security | Underwater sensor network |
| Rechargeable seawater-activated batteries | Anode: $Na \leftrightarrow Na^+ + e^-$ <br> Cathode: $O_2 + 2H_2O + 4e^- \leftrightarrow 4OH^-$ | | Rechargeable; Long cycle lifespan; Low cost | Diverse marine sectors; typical energy storage applications |

The various types of seawater-activated batteries can be classified according to the function of seawater; these include high-power power batteries for underwater weaponry, long-cycle low-power DO batteries for underwater detection instruments, metal semi-fuel batteries for UUVs, and rechargeable batteries for energy storage systems. As all these batteries employ seawater as an electrolyte to some extent, they are considered seawater-activated batteries. Nevertheless, they are fundamentally different in terms of their structures, electrical properties, and principles. In this review, we discuss the principles, structures, and characteristics of three different types of seawater-activated batteries and summarize the research progress, application prospects, and future trends in the development of seawater-activated batteries. Few articles in the published literature describe seawater-activated batteries, and this review provides a comprehensive introduction to seawater-activated batteries. This review displays past research routes and the relevance of the on-going study to previous research work. Furthermore, this review summarizes some of the research results in the field of seawater-activated batteries giving an insight into possible further research directions.

## 2. Metal Semi-Fuel Seawater-Activated Batteries

### 2.1. Dissolved Oxygen (DO) Seawater Batteries (DO-type Seawater Batteries)

Small-power DO-type seawater batteries generally use an Mg or Al alloy as the negative electrode material; the positive reaction substance is the DO in seawater and the electrolyte is natural seawater. The standard electrode potential and Faraday capacity of Mg are –2.38 V and 2200 mAh/g, respectively, while those of Al are –1.67 V and 2980 mAh/g, respectively. These values are much higher than for Zn, and the prices of Mg and Al are low, making them good battery anode materials. The positive electrode also has a high catalytic capacity for oxygen reduction [40].

The main role of natural seawater as a battery electrolyte Is to provide a medium for the directional movement of ions, thereby forming a stable current (Figure 2). The flow of seawater can facilitate oxygen transport while washing away the reaction products. Thus, Electrochemical behavior are designed with an open structure, which is safer in the marine environment, especially in the deep sea [41,42]. The electrochemical reactions of DO-type seawater batteries are as follows:

$$\text{Anode: } M \rightarrow M^{n+} + ne^-,$$

$$\text{Cathode: } O_2 + 2H_2O + 4e^- \rightarrow 4OH^-.$$

During the anodic half-reaction, metals generally undergo self-corrosion, such that:

$$M + nH_2O \rightarrow M^{n+} + nOH^- + (n/2)H_2.$$

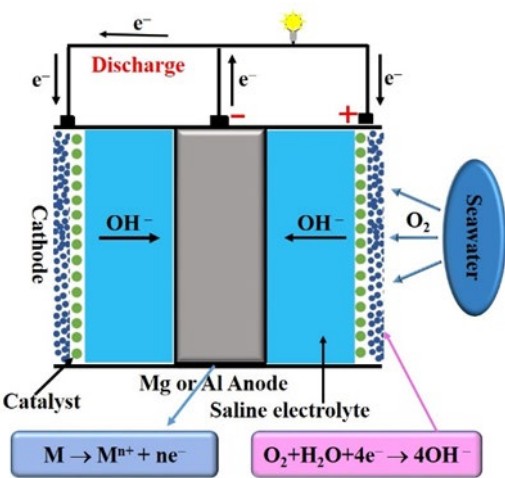

**Figure 2.** Reaction in dissolved oxygen (DO)-type seawater batteries.

Zhang and Yang et al. prepared an oxygenophilic atomic dispersed Fe–N–C catalyst for lean-oxygen seawater-activated batteries. The Fe–N–C/CNT catalyst displayed an excellent ORR performance in both $O_2$-saturated alkaline medium and neutral seawater with half-wave potential of 0.920 V and 0.704 V, respectively, much better than the commercial Pt/C catalyst. Seawater-activated batteries presented a remarkable performance in oxygen-poor seawater (ca.0.4mg/L), with a discharge voltage of 1.18 V at the current density of 10 mA/cm² [43].

Owing to the low concentration of oxygen in seawater, the output power of DO-type seawater-activated batteries is small; consequently, their use is generally limited to beacon lights, life-saving equipment, and small marine detection instruments. Because the battery reaction process only requires the consumption of Al and Mg alloys as raw materials, it is a very economical and environmentally friendly means of providing energy for marine instrumentation. In 1996, the automatic control system of a submarine oil well located at a depth of 180 m in the Ionian Sea used six 2-m-high seawater batteries for its power supply. The negative electrode of the battery was a commercial Mg alloy, the positive electrode was an inert carbon fiber bundle, and the DO from seawater was used as the oxidizer. The battery had an open structure and was used stably without failure, demonstrating the safety and applicability of DO-type seawater batteries for marine exploration [44].

In 2000, Japan established a 5500 m ocean borehole seismic observation platform in the northeastern Pacific Ocean, which is powered by a seawater-activated battery system consisting of four SWB1200 batteries connected in parallel. The open-cell frame of this system is composed of high-density Ti alloy, with the positive electrode being a carbon fiber bundle connected in parallel and the negative electrode being a commercial Mg alloy that is replaced by a diving robot when depleted to ensure a continuous energy supply. This battery system was able to steadily provide power for up to five years, with an average power of 13 W and a high energy density. This application illustrates the long–term suitability of seawater batteries in deep-sea environments [45].

Based on the structural design of the commercial seawater battery SWB1200, Xu et al. [46] developed an oxidized carbon fiber brush (CFB) as the anode material of a battery using titanium wire and a carbon fiber bundle as raw materials and then conducted three

real-world prototype tests at sea. The CFB electrode exhibited quasi-capacitance characteristics in natural seawater; therefore, the battery was designated as a seawater supercapacitor DO battery. Cyclic voltammetry and steady-state constant-current discharge methods were used to study the discharge performance of the oxidized CFB and Mg alloy, and the results showed that the seawater battery had a volumetric specific power density of 5.8 mW/L, which is better than that of the SWB1200 battery, and demonstrates its suitability for long-term energy supply of medium-power instrumentation.

The Directorate General of Armame"ts/B'ssin d'Essais des Carenés of France and Norwegian Defence Research Establishment (FFI) jointly developed the CLIPPER AUV. The vehicle was powered by DO-type seawater batteries, with six seawater batteries distributed around a central pressure-resistant housing. Each seawater cell had 234 AZ31B magnesium rods as the negative electrode and 190 parallel carbon fiber bundles as the positive electrode. The total power of the battery pack was 678 W when the vehicle was moving at a speed of 2 m/s, which was sufficient to support it for two weeks near the Arctic at a depth of 600 m. Notably, seawater batteries are extremely safe because they do not require a specific pressure-resistant container for protection [47].

### 2.2. Metal Hydrogen Peroxide Seawater Batteries

A metal semi-fuel cell with either Mg metal or Al as fuel and $H_2O_2$ as an oxidizer is a new type of underwater chemical power source that has been developed. This type of battery has been used in data acquisition and other electronic power-supply instruments. With the increasing demand for new high-energy power sources for marine resource development, defense, and ecological research, such batteries have been rapidly developed in recent years [48]. The electrochemical reactions of metal hydrogen peroxide seawater batteries are as follows (Figure 3):

$$\text{Anode: } 2Al + 8OH^- = 2Al(OH)_4^- + 6e^-,$$

$$\text{Cathode: } 3H_2O_2 + 6e^- = 6OH^-,$$

$$\text{Battery: } 2Al + 3H_2O_2 + 2OH^- = 2Al(OH)_4^-.$$

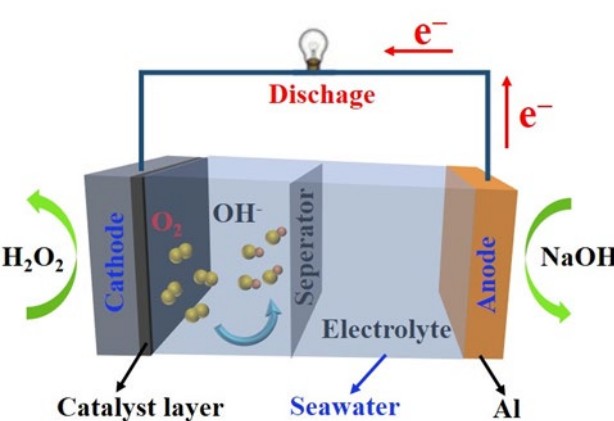

**Figure 3.** Reactions in metal semi-fuel seawater batteries.

Bessette et al. investigated $Ir_2O_3$–Pd core-shell structural catalyst synthesized by the electrodeposition method, which was then loaded on active carbon and used as the cathode of the Al–$H_2O_2$ semi-fuel cell [49]. Sun et al. reported the Mg–$H_2O_2$ fuel cell using Pd-Ag/Ni foam as the electrocatalytic cathode reached a power density of 80 mW/cm$^2$ at 25 °C [50]. Then, his research group also prepared a palladium cathode onto the titanium

foam for the Mg–H$_2$O$_2$ fuel cell and the power density of this fuel cell (110 mW/cm$^2$) was much higher than that of the Pd–Ag/Ni cathode.

In the late 1960s, Zaromb et al. first proposed a metal peroxide cell with Al and Mg alloys as the anode, H$_2$O$_2$ as the active cathode material, and a hydrogen peroxide reduction catalyst as the cathode fluid. Dow et al. [51] investigated the use of an isolation membrane between the anode and cathode of an Al–H$_2$O$_2$ semi-fuel cell, in which the anode was an Al alloy, and the cathode was a three-dimensional porous mesh made of carbon fibers with Pd/Ir as the H$_2$O$_2$ reduction catalyst. The electrochemical efficiency of this diaphragm-type semi-fuel cell was 75% higher than that of the membrane-type cell.

Aluminum–H$_2$O$_2$ semi-fuel cell was first developed by Hasvold et al. [52] in 1998 and used in the Hugin 3000 UAV, in which it was able to extend the endurance up to 60 h and outperformed lithium-ion batteries. An Al–H$_2$O$_2$ semi-fuel cell was developed and successfully used by the Norwegian FFI in response to AUV requirements. This battery has a high specific energy of 260–400 Wh/kg more than 10 times that of a lead-acid battery and more than three times that of a zinc-silver battery. Therefore, Norway, Canada, and the United States have used Al–H$_2$O$_2$ rather than zinc-silver batteries to extend the endurance of AUVs from 4–10 h to 30–60 h.

Medeiros et al. [53] studied Mg–H$_2$O$_2$ semi–fuel cells with an anode of Mg alloy AZ61, a conductive ion membrane of propylene glycol-treated Nafion-115, and a cathode of carbon-fiber-loaded Pd–Ir. Both the anode seawater and cathode electrolytes were seawater. A single cell could be discharged continuously for 30 h with a current density of 25 mA/cm$^2$, a stable voltage between 1.77 and 1.8 V, and a specific energy of 500–520 Wh/kg.

### 3. High-Power Seawater-Activated Batteries

Negative high-power seawater-activated battery electrodes are mainly produced using Mg or Al alloy, and the positive electrode is mainly based on silver, lead, and copper oxide series or chlorides, which are suitable for high-current discharges. The working principle of high-power seawater-activated batteries is the same as that of primary batteries, with the electric potential generated by the difference in potential between the positive and negative electrodes of different materials, which causes the ions in the electrolyte to move directionally to form a current (Figure 4). These batteries are also safe, reliable, and exhibit high low–temperature performance, high specific energy and power, and long storage lives [54].

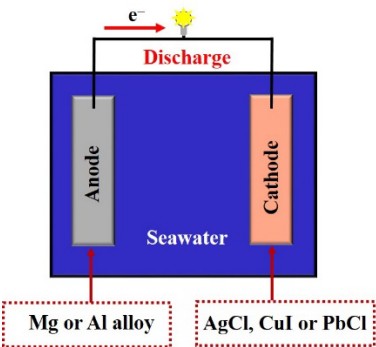

**Figure 4.** Reactions in high-power seawater-activated batteries.

The most successful application of high-power density batteries is in power batteries for underwater torpedoes (Figure 5). As the power and energy density of batteries used in modern torpedoes must be high, power batteries with natural seawater as the electrolyte are commonly used because they can effectively reduce the weight carried by the torpedo and thus significantly increase the energy density of the battery. To date, Mg/AgCl, Mg/CuCl, and Al/AgO seawater batteries have proven practical (Table 2); these batteries are different from older low-power DO-type seawater batteries, generally using

a lower electrode potential and more active metal materials as the anode and the circulation of natural seawater as the electrolyte to exclude the products of the electrode reaction and prevent surface polarization. They can also remove part of the electrode reaction process, some of the heat released from that process, and control the temperature of the battery [55,56].

He et al. [57] reported a high-purity AgO cathode material for superior performance Al-AgO seawater-activated batteries. The synthesized AgO cathode exhibited an outstanding specific capacity of 394.7 mAh/g, and the discharge voltage plateau is closed to 1.7 V at the current density of 500 mA/cm$^2$, with excellent charge transfer kinetics and electronic conductivity. Moreover, the maximum output power density of 1260 mW/cm$^2$ at the current density of 900 mA/cm$^2$ is also possible. Kong et al. [58] reported a study of the discharge/charge process of a new AgCl/Ag/Carbon felt composite electrode for seawater–activated batteries. In their report, the prepared AgCl/Ag/Carbon felt composite cathode showed a high original discharge/charge capacity of 179.0 mAh/g and 223 mAh/g, individually. In the third cycle, the discharge/charge specific capacities of AgCl/Ag/Carbon felt composite cathode was maintained at 90% and 86%, respectively.

**Table 2.** Reactions in high-power seawater-activated batteries with different anode and cathode materials.

| Type | Mg/AgCl | Mg/CuCl | Al/AgO |
|---|---|---|---|
| Anode | $Al + 4OH^- \rightarrow Al(OH)_4^- + 3e^-$<br>$AgO + H_2O + 2e^- \rightarrow Ag_2O + 2e^-$ | $Mg \rightarrow Mg^{2+} + 2e^-$ | $Mg \rightarrow Mg^{2+} + 2e^-$ |
| Cathode | $Ag_2O + H_2O + 2e^- \rightarrow 2Ag + 2OH^-$ | $CuCl + 2e^- \rightarrow Cu + Cl^-$ | $AgCl + e^- \rightarrow Ag + Cl^-$ |

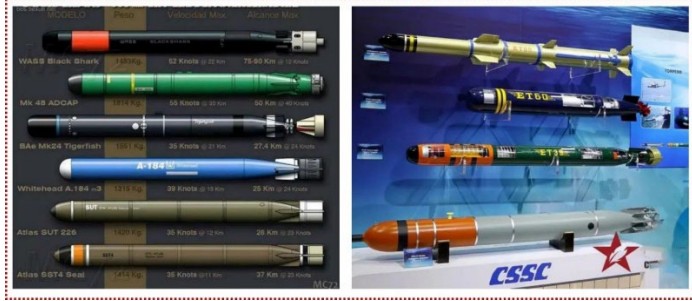

**Figure 5.** Examples of torpedoes successfully developed in China and abroad.

Magnesium/AgO batteries have been the most widely used in seawater-activated batteries since their development in 1940 by Bell Laboratories. However, because of the expensive materials used in the cathode, the cost of these batteries remains high, so they are generally used as power batteries for torpedoes. For example, the Italian A244, American MK44, and British Sting Ray torpedoes used Mg/AgO seawater batteries. Torpedoes propelled by this generation of batteries can generally reach speeds of 30–40 kn (1 kn is equal to 1.852 km/h or 0.514 m/s) and a range of approximately 20 km. The British Sting Ray light torpedo uses a 63 kW Mg/AgO battery pack and can reach a speed of 45 kn (range: 30–40 kn) and a range of 20 km [59].

The United States Navy began to research Al/AgO batteries in the 1970s, and this became an important milestone in the development of torpedo power batteries. Such batteries are suitable for high-current discharge and have working current densities of approximately 600–1000 mA/cm$^2$ (i.e., 3–5 times the current density of zinc-silver batteries). The cell monomer voltage is between 1.6 and 1.7 V and is higher than those of both zinc-silver and Mg/AgO cells. The mass-specific power of Al/AgO batteries is 130 Wh/kg and the volume-specific power ranges up to 2000 W/dm$^3$. Examples of Al/AgO batteries used as power sources for light torpedoes are the Italian A290, French "Sea Eel", and the

French-Italian MU90 torpedo. The "Sea Eel" has a maximum speed of 53 kn and a minimum speed of 38 kn; the A290 torpedo, which was developed at a similar time as the "Sea Eel" can reach a maximum speed of 57 kn and is currently the most advanced rocket-assisted torpedo. After the successful application of Al/AgO batteries in light torpedoes, the United States and some European countries began researching their use as power sources for heavy torpedoes. The "Black Shark" heavy torpedo developed by the Italian Whitehead Company uses an Al/AgO battery and has a maximum range of more than 50 kn [60–62].

Magnesium/CuCl seawater batteries were developed in the former Soviet Union based on a Mg/AgO battery and they are still widely used in Russia. The cathode of this type of battery uses a relatively economical Cu alloy material instead of expensive metallic silver and copper chloride as the anode. To prevent oxidation of the CuCl cathode material, a certain amount of $SnCl_2$ is added, whereas hydrogen protection measures are used to ensure the activity of the electrode. Magnesium amalgam is used as the anode in these batteries and amalgamation improves the stability of the magnesium and the surficial hydrogen precipitation overpotential, inhibiting the self-corrosion of the magnesium anode. The cost of this type of battery is one-third that of an Al/AgO battery, but its large size required an increase in the length of the 53 K light torpedo and improvements to the control system. Therefore, it is only used in Russia for torpedoes such as the TC-3 and T-80.

## 4. Rechargeable Seawater-Activated Batteries

Rechargeable seawater-activated batteries that use seawater as an unlimited resource and source of $Na^+$ ion cathodes for energy storage have recently been developed. In these batteries, the anode is protected from the seawater cathode by a $Na^+$ ion-conductive solid electrolyte, which can block water molecules and ions other than $Na^+$ (Figure 6). Rechargeable seawater-activated batteries have an open-structure cathode system in which seawater is usually exposed to the ambient air. The anode stores the $Na^+$ ions harvested from the seawater by charging; therefore, the anode plays a vital role in determining the energy of rechargeable seawater-activated batteries [39,63,64].

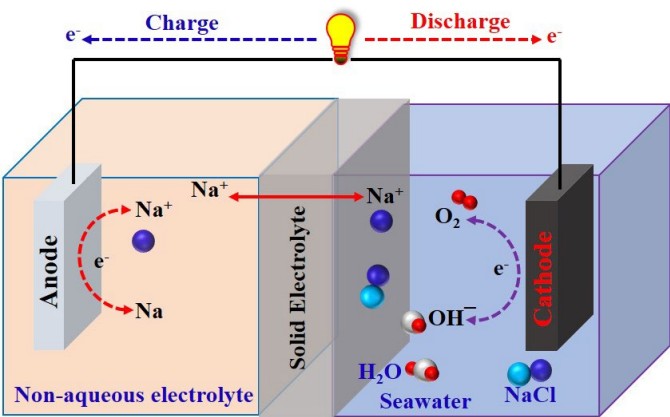

**Figure 6.** Diagram of a rechargeable seawater-activated battery in the charged-discharged state.

The electrical energy in rechargeable seawater-activated batteries is stored at the anode side as Na metal by harvesting $Na^+$ ions from seawater via the oxygen evolution reaction (OER) during the charging process. During the discharge process, the easily reducible species of DO present in seawater are involved in the oxygen reduction reaction (ORR) with the aid of water. The stored chemical energy is then released as electricity with $Na^+$ ions transferred back from the anode into the seawater. The half-cell (anodic and cathodic) reactions are as follows:

Cathode (in seawater, pH = 8): $4OH^- \leftrightarrow O_2 + 2H_2O + 4e^-$ (ORR $\leftrightarrow$ OER)

E = 0.77V vs. SHE (Standard Hydrogen Electrode)

Anode (in a nonaqueous electrolyte): $4Na^+ + 4e^- \leftrightarrow 4Na$

E = –2.71V vs. SHE

Overall reaction: $4Na + O_2 + 2H_2O \leftrightarrow 4NaOH$

E = 3.48V vs. $Na/Na^+$

Currently, rechargeable seawater-activated batteries have attracted massive attention from researchers. Jung et al. reported the sodium metal anode by use of pre-patterned current collector for rechargeable seawater-activated batteries [65]. They found that the sodium ions are priorly deposited on the surface of the metal with correspondingly large binding energy which allow us to create a patterned coating on the current collector. Additionally, Kim et al. studied a promising anode material for rechargeable metal-free seawater-activated batteries. Their group prepared an amorphous red phosphorus/carbon composite anode material, which is successfully used as an anode for seawater-activated batteries. It presented stable cycling performance with a reversible capacity of over 920 mAh/g and a coulombic efficiency of over 92% in 80 cycles and a good rate capability [66].

Rechargeable seawater-activated batteries can be operated as an open cathode system, where the charge and discharge processes are completed in the seawater. Therefore, these batteries are expected to be more suitable for the marine sector, offshore and seaside power sources, and energy storage applications than other types of batteries [67–70]. The possible applications of rechargeable seawater-activated batteries are shown in Figure 7.

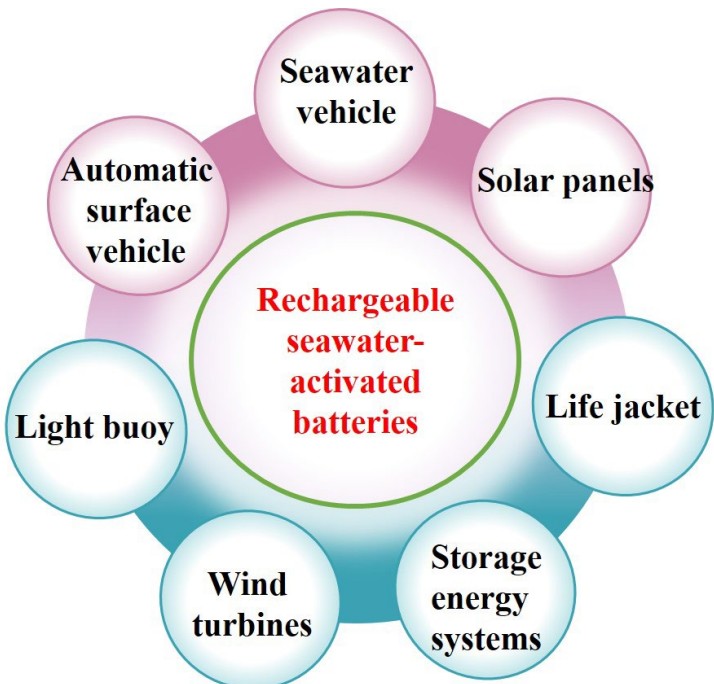

**Figure 7.** Potential applications for rechargeable seawater-activated batteries.

Batteries used in life-jacket applications should provide the instant power necessary to activate the mounted Global Positioning System (GPS) and light source(s). The anode of rechargeable seawater-activated batteries is completely isolated from the cathode so that when the cathode comes into contact with seawater, it can provide instant power. Therefore, marine life jackets can use this type of battery (in the charged state) to power a

light source and operate the GPS, the latter of which can provide the position of a person who needs help in emergencies (e.g., following a shipwreck or when a boat capsizes).

A light buoy is an illuminated object, such as a lighthouse, that is used along the coast and in the open ocean to aid ships and boats in navigation and to warn of obstructions. Current light buoys mostly use toxic, low-energy-density, and heavy lead-acid batteries to power mounted light-emitting diode lights, GPS, and monitoring systems (used for monitoring water temperature, current/wind speed and direction, and salinity). Compared with lead-acid batteries, rechargeable seawater-activated batteries can provide a high-energy-density, eco-friendly, low-weight, and maintenance-free option. Therefore, it is possible to replace lead-acid batteries with rechargeable seawater-activated batteries in light buoys.

Autonomous Surface Vehicles (ASVs) are similar to floating equipment in seawater, such as light buoys. They are used to evaluate water depth and detect obstructions using ultrasonic signals. Most ASVs are deployed in ports and harbors. Rechargeable seawater-activated batteries can store power from solar panels and provide a constant power supply for 24 h of ASV operation.

Remotely operated vehicles (ROVs), such as underwater robots and drones, are used by the military and heavy industries (e.g., oil companies), as well as for oceanographic and marine biological research. Underwater scooters (UWSs) are used for tours to view colorful and intricate reefs. As all ROVs and UWSs are operated by an electrical power supply of batteries, and rechargeable seawater-activated batteries are good candidates for adoption as power sources in ROVs and UWSs.

Nearly 50% of the world's population lives by the sea. Of the 17 largest cities in the world, 14 are located near the coast, and they consume most of the electricity. The required electricity is transmitted from long distances, increasing costs and reducing energy efficiency. Recently, renewable energy technologies such as solar, wind, wave, and tidal energy generators have been deployed offshore and by the sea to reduce energy transmission distances. However, large-scale EES systems are necessary to provide stable electrical energy, and rechargeable seawater-activated batteries can be a better option for large amounts of electrical energy storage. It is expected that offshore and waterfront deployments of rechargeable seawater-activated batteries may be very easy, since they operate in seawater as the main battery component [71,72].

## 5. Directions for Future Development

First, the performance of electrode materials should be optimized. Currently, Mg and Al alloys are the most commonly used anode materials in seawater-activated batteries. The optimization of material properties by alloying has achieved clear results, but there is still room for improvement. While the alloying elements used Li, Zn, Ga, and In, there is no uniform explanation of their mechanism(s) of action; moreover, the number of alloying elements is expanding as researchers learn more about metal elements. Previous research has shown that the introduction of a certain amount of rare earth elements into the anode material can refine the microstructure of the material, further improving the efficiency of the Mg and Al anodes. In addition to Al and Mg metals, other metals may be used to create anodes in the future. Secondly, the electrolyte control system must be optimized. As seawater-activated batteries use natural seawater, the nature of that water (temperature, salinity, and flow rate) significantly impacts battery performance. Our understanding of the movement of ions in electrolytes remains lacking. Therefore, it is necessary to control and optimize the electrolyte system and doing so should greatly improve battery performance. Third, the kinetic mechanism(s) of the electrochemical reactions of seawater-activated batteries remain(s) unclear. At present, single electrode performance is mainly studied using a constant current or voltage method with a three-electrode electrochemical system. However, electrode performance in this state differs from the electrode polarization characteristics in an actual battery. Therefore, the results of such studies can-

not be simply used to replace the electrode performance in an actual battery. In situ synchronous tracking methods for microscopic electrochemical reaction processes on the surface of electrode materials are also lacking, such that the study of battery performance remains focused on macroscopic performance indices, such as charge and discharge performance. These drawbacks affect the kinetics of the electrode-reaction process. Finally, the overall control system of seawater-activated batteries should be optimized. The research and development of batteries is systematic and cannot be limited to the research of electrode materials, electrolytes, and charge/discharge performance, but should also include the temperature control, gas–liquid separation, inlet and outlet valve, and electrical control systems.

## 6. Conclusions

Seawater-activated batteries, which utilize the physical and chemical properties of seawater to achieve a flexible, distributed, in situ power supply have broad application prospects. Depending on the purpose of use, seawater-activated batteries can be divided into three categories, ranging from those that have a long life but low power output to those with a high current and high power, indicating that seawater-activated batteries may be widely adapted. In recent years, with in-depth research on energy technologies and electrode materials, such batteries have attracted renewed attention and have continued research and development value, especially for use in UUVs and torpedoes. Therefore, they are urgently needed and warrant intensive research.

**Author Contributions:** Conceptualization, W.X. and X.W.; validation, C.X., W.X., and X.W.; formal analysis, S.Y.; investigation, S.Y.; resources, J.C.; data curation, W.X.; writing—original draft preparation, S.Y.; writing—review and editing, W.X.; visualization, X.W.; supervision, C.X.; project administration, W.X.; funding acquisition, W.X. All authors have read and agreed to the published version of the manuscript.

**Funding:** This research received no external funding.

**Institutional Review Board Statement:** Not applicable.

**Informed Consent Statement:** Not applicable.

**Data Availability Statement:** Not applicable.

**Acknowledgments:** All the authors thank the Institute of System Engineering, Academy of Military Science for their support to our work., All the authors thank the reviewers for their valuable comments on our manuscript.

**Conflicts of Interest:** The authors declare no conflict of interest.

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
