# Peer review of "Progress and Applications of Seawater-Activated Batteries"

_sustainability, doi:10.3390/su15021635_

Round 1

Reviewer 1 Report

This transcript provides a review on to seawater activated batteries. The authors classified the batteries into several types and provided an outlook on the future of seawater activated batteries.

Overall, the transcript reads very well. The authors did a good job introducing the subject. However, I have some comments on it:

1- In the introduction (section 1), I strongly suggest adding a couple of sentences in the last paragraph emphasizing the contribution this paper provides in this field. Review papers should always have a clear and clean contribution. 

2- Could you define DO-type batteries in the text? You should add that to the introduction in paragraph 2.

3- It is recommended to add how much 1 kn is in km/h or m/s so the reader can relate.

4- The text in figure 7 needs to be a bit larger. (roughly the size of the text).

5- Sustainability journal has several recent publications that are related to different types of batteries (in general). Some of them can fit very well with the context of the introduction here like (to name a few):

https://doi.org/10.3390/su142114169

https://doi.org/10.3390/su142113708

https://doi.org/10.3390/su142013417

Author Response

Dear Reviewer:

On behalf of my co-authors, we thank you very much for allowing us to revise our manuscript, we appreciate the editor and reviewers very much for their positive and constructive comments and suggestions on our manuscript entitled “Progress and Applications of Seawater-Activated Batteries”.(ID: sustainability-2047497)

Those comments are all valuable and very helpful for revising and improving our paper, as well as the important guiding significance to our research. We have studied the comments carefully and have made a correction which we hope meet with approval. Revised portions are marked in red on the paper.

Responds to the reviewer’s comments:

Comments:

This transcript provides a review on to seawater activated batteries. The authors classified the batteries into several types and provided an outlook on the future of seawater activated batteries.

Overall, the transcript reads very well. The authors did a good job introducing the subject. However, I have some comments on it:

Point 1: In the introduction (section 1), I strongly suggest adding a couple of sentences in the last paragraph emphasizing the contribution this paper provides in this field. Review papers should always have a clear and clean contribution. 

Response 1: Thanks a lot for your suggestion. We completely agree with you that adding a couple of sentences in the last paragraph emphasizes the contribution this paper provides in this field. Few articles in the published literature describe seawater-activated batteries, and this review provides a comprehensive introduction to seawater-activated batteries. This review displays past research routes and the relevance of the ongoing study to previous research work. Furthermore, this review summarizes some of the research results in the field of seawater-activated batteries giving an insight into possible further research directions.

Point 2:Could you define DO-type batteries in the text? You should add that to the introduction in paragraph 2.

Response 2: I am very grateful for your comments on the defined DO-type batteries. According with your advice, we carefully explain DO-type batteries in paragraph 2. Dissolved oxygen seawater-activated batteries (DO-type seawater batteries) use magnesium/aluminum as the anode, carbon as the cathode, seawater as the electrolyte, and dissolved oxygen in seawater as the oxidant.

Point 3: It is recommended to add how much 1 kn is in km/h or m/s so the reader can relate.

Response 3: Thanks to you for your good comment about explaining the correspondence between 1 kn and km/h. The 1kn is equal to 1.852 km/h or 0.514 m/s.

Point 4: The text in figure 7 needs to be a bit larger. (roughly the size of the text).

Response 4: Thanks a lot for pointing this out. We have modified the size of the text in Figure 7 in the revised manuscript.

Point 5: Sustainability journal has several recent publications that are related to different types of batteries (in general). Some of them can fit very well with the context of the introduction here like (to name a few):

https://doi.org/10.3390/su142114169

https://doi.org/10.3390/su142113708

https://doi.org/10.3390/su142013417

Response 5: Thank you very much for your good advice, we have carefully read the references you have provided, and these references are indeed very valuable in the introduction to this review, so we cite the Ref.5, Ref.6, Ref.8 in the first paragraph of the introduction, respectively.

Reviewer 2 Report

This review provides a comprehensive introduction to seawater-activated batteries. Here, They classify seawater- activated batteries into metal semi-fuel, high-power, and rechargeable batteries according to the different functions of seawater within them. The working principles and characteristics of these batteries are then introduced, and we describe their research statuses and practical applications.

1Authors said development about electrode materials in section 5,  while there is no description about electrode materials through the whole paper. Add some electrode material performancelike charge/discharge performance.

2There is no performance comparison of different batteries. Authors should revise this.

3Authors should improve the resolution of all the pictures in the manuscript, like fig.7 which is too small to read.

4If the authors have got the copyright of the pictures insert in fig.7.

5The current manuscript mainly explains the working mechanism of the batteries but lacks the description of the research status. 
